# Experimental Investigation on the Bonding Strength of Knotted CFRP Bars in Bulk Plastics

**DOI:** 10.3390/polym15092036

**Published:** 2023-04-25

**Authors:** Cihan Ciftci

**Affiliations:** 1Department of Civil Engineering, Abdullah Gul University, 38080 Kayseri, Turkey; cihan.ciftci@agu.edu.tr; 2Techno-CC R&D Innovation Co., Ltd., Erciyes Technology Development Zone, 38080 Kayseri, Turkey

**Keywords:** sliding failure, bonding strength, reinforced plastics, composite structures, carbon fiber reinforced polymers

## Abstract

Improving the interfacial bonding strength of CFRP materials is crucial for enabling the development of novel composite beam structures with higher specific bending strength demanded by the composite industry. In this research study, for reinforced bulk plastic composites, the aim is to enhance the interfacial bonding strength of CFRP bar elements in bulk plastics by on the formation of knots. In this context, firstly, the knotted CFRP bars with varying cross-sectional areas were manufactured under laboratory conditions for the experimental investigation on the effect of knots on bonding strength. Commercially available smooth-surfaced CFRP bars were also purchased to be used as the reference. Then, all these CFRP bars were subjected to pull-out tests by using in bulk plastics. According to the test results, it was observed that the interfacial bonding strength of CFRP bars in bulk plastic materials could be increased up to 233% because of the knots.

## 1. Introduction

Composite structures containing carbon-based reinforcements are frequently preferred by aviation, aerospace, energy and automotive industries due to their robust specific strength and stiffness [1,2]. The usage rate of the composite structures in these industries has been further increased in time owing to the development of technology and decreasing costs. For example, by the latest technological developments, the amount of composite structures has been maximized up to about 50% of the aircrafts by weight with the production of Boeing-787 Classics [3]. However, the remaining structural elements in this Classic still consist of metals at high rates (steel 10%, titanium 15%, and aluminum 20%) [3]. The continuation of increasing the usage rate of the composites requires increasing the strength of the carbon-based composite structures and reducing costs. To meet this requirement, innovative developments of composite structures with different strength properties and low costs are ongoing from the past to the present. For instance, several significant composite structures such as foam core sandwich beams (e.g., [4,5]), honeycomb core sandwich panels (e.g., [6,7]), corrugated core sandwiches (e.g., [8,9,10,11,12]), and lattice truss core sandwiches (e.g., [13,14,15,16,17]) were developed and also improved by researchers. It is also worth noting that these researchers have recently focused more on lattice truss core sandwich structures than the others due to their higher specific bending strength capacity. In addition to all these developed composite structures, according to the author, it may be possible to develop a new and worthwhile carbon-based composite structure by conceptually utilizing the design of reinforced concrete structures, which is commonly used in civil engineering constructions. In other words, high-strength, low-cost, and lightweight composite structures can be constructed by replacing steel reinforcements and concrete material, which is used in reinforced concrete structures, with carbon-based reinforcing bars and bulk plastic materials, respectively. For example, in this regard, carbon reinforced bulk plastic (CRBP) beams were produced in 2019 using 3D printers [18]. According to the experimental data of the research study [18], the specific bending strength of the CRBP beams was found to be 0.37 (288 MPa/786 kg/m^3^). Therefore, it can be stated that the specific bending strength of the CRBP beams is comparable to the specific bending strength (e.g., 0.26 MPa/kg/m^3^ in [17]) of the innovative truss core sandwich composite beams.

It is anticipated that having a high bending strength for a CRBP (carbon reinforced bulk plastic) beam depends on two main conditions, as in the concept of the design of reinforced concrete structures [19]. These are (1) to have adequate strength capacity of the reinforcements in bulk plastics, and (2) to have adequate bonding strength capacity of the reinforcements in bulk plastics. For the reinforced bulk plastics, the bending failure type is probably related to the latter one, because it is already known from numerous studies (e.g., [20,21,22]) in the literature that the CFRP composite structures have robust strength and stiffness. For the secondary condition, any sliding movement that may occur between the bulk plastic material and the reinforcing carbon bars will cause the structural system to fail and even cause a sudden collapse. 

To the best of the author’s knowledge, the bonding strength of the carbon-based reinforcements in bulk plastics has not been examined in a research study. On the other hand, in the literature, there are numerous studies (e.g., [23,24,25,26,27,28,29]) focusing on slipping failures of carbon or glass fiber reinforcements in concrete, not in bulk plastics. The main motivation of these works is not to meet the needs of the aforementioned sectors, but to prolong the service life of the reinforced concrete structures by replacing steel reinforcements with the fiber materials. The reason for this replacement is that the carbon or glass fiber reinforced polymer bars have a much higher resistance against the corrosion than the steel bars [24,30,31]. The most important problem of these fiber reinforced concrete structures is to have insufficient bonding strength of the reinforcements in concrete [25,30,31]. In order to increase this bonding strength, a number of research studies (e.g., [23,25,27,32,33,34]) have been carried out by civil engineers, especially over the last two decades. In these studies, several techniques such as sanded coating, grooving, and helical wrappings on the surface of bars made of fiber materials were applied in order to increase the bonding strength of the reinforcements in concrete. Despite all these studies, the bonding strength of fiber reinforcements in concrete has not fully reached the desired values [30,35,36]. Thus, unfortunately, the use of carbon or glass fiber reinforcing bars in concrete is not widely preferred by the construction industry, at present. Although the technological progress in the bonding strength of continuous CFRP bars in concrete is not enough as desired, some researchers (e.g., [37,38]) have also studied on the strength properties of fiber reinforced cementitious composite materials.

The main motivation of this study is to pave the way to produce new lightweight and higher-strength composite structures by minimizing the slipping failures related to the interfacial bonding strength between the reinforcements and bulk plastics. This is because it is thought that increasing the bending strength of reinforced bulk plastics is based on increasing the interfacial bonding strength between the reinforcements and the plastic materials. For this purpose, as the novelty of this study, a new application has been developed that will make the carbon-based reinforcements difficult to slip in bulk plastics by forming knots on them. The positive effect of this developed application on the interfacial bonding strength between the reinforcements and the plastic materials was also experimentally demonstrated in this study. According to these experimental results, the bonding strength of the reinforcements in the bulk plastic material could be increased up to 233%.

## 2. Materials and Methods

Raw materials (carbon fiber bundles and resin components) were purchased from a company in Istanbul, Turkey (Dost Kimya Industrial Raw Materials & Trade Co., Ltd., Istanbul, Turkey). According to the company, the tensile strength, elongation capacity and modulus of elasticity of the fiber bundles are 2.3 GPa, 1.8 % and 130 GPa, respectively. Additionally, the resin material (MGS LR285-Hexion) was used with the hardener component (MSG LH287-Hexion).

First, all the samples produced within this study are addressed under the sub-section of “sample preparation”. In the other sub-section, the produced samples are investigated under pull out tests in order to see the bonding performance of the CFRP bars in bulk plastics.

### 2.1. Sample Preparation

The knotted CFRP bars of samples were firstly produced in laboratory using the manufacturing process represented in Figure 1. In this figure, an impregnation pultrusion system is described. According to this description, at first, the required number of fiber bundles are impregnated in a resin bath. At the second step, the impregnated fiber bundles are knotted. After cutting off the knotted bundles, they are hanged under a mass-load at the curing stage, both to prevent slackening of the knots on the wet bundles and to obtain a straight bar.

In order to experimentally examine and compare the bonding characteristics of FRP bars with and without knotted states, a total of 6 sample sets were prepared as shown in Table 1. The main difference between these sample sets is based on to use different FRP bars in each sample set. As in Table 1, the names of these sample sets are designated corresponding to the types of CFRP bars embedded in the sets. The designations of these sample sets are: (1) CA-SS-CFRP: samples containing commercially available (CA) smooth surfaced (SS) CFRP bars; (2) LM-FS-8B-CFRP: samples containing lab-made (LM) flat surfaced (FS) CFRP bars created by amalgamating 8 unidirectional fiber fabric bundles (8B) with epoxy; (3) LM-K-2B-CFRP: samples containing lab-made (LM) knotted (K) CFRP bars created by amalgamating 2 unidirectional fiber fabric bundles (2B) with epoxy; (4) LM-K-4B-CFRP: samples containing lab-made knotted CFRP bars created by amalgamating 4 unidirectional fiber fabric bundles (4B) with epoxy; (5) LM-K-6B-CFRP: samples containing lab-made knotted CFRP bars created by amalgamating 6 unidirectional fiber fabric bundles (6B) with epoxy; and (6) LM-K-8B-CFRP: samples containing lab-made knotted CFRP bars created by amalgamating 8 unidirectional fiber fabric bundles (8B) with epoxy, respectively. The preparation of all the sample sets was repeated in 3 times, thus a total of 18 samples were prepared and tested in this study. Figure 2 has several selected examples to show the final state of the samples after the preparation. The samples, which are in the 1st, 2nd, and 6th sets, were produced with a similar cross-sectional area (diameter is approximately 4 mm) in order to compare the effect of the structural forms (with or without the knots) of the CFRP bars on their bonding properties with regardless size effect. Additionally, to compare the size effect of the knotted CFRP bars on their bonding performance, the samples, which are in the 3rd, 4th, 5th, and 6th sample sets, were produced and tested.

To create the knotted FRP bars used in the sample sets, several steps are followed. First, unidirectional carbon fiber fabric bundles are impregnated in a resin pool. Second, the impregnated bundles are longitudinally amalgamated to each other. Third, the amalgamated and bendable fabric bundles are tied to form a knot in different locations as shown in Table 1. Fourth, the knotted and impregnated fabrics are subjected to tensile forces during the curing of the resin material to form a stiff bar. Furthermore, the locations of the knots are important. In other words, the location of the four knots should coincide with the 18 cm long bulk plastic zone, whereas the fifth one should be approximately at the midpoint of the 3 cm long bulk plastic zone (see Figure 2). The main reason for the usage of more knots in the 18 cm long bulk plastic zone is to prevent any slippage that may occur in this zone while testing, and also to ensure that all the slippages occur in the 3 cm long zone. 

After producing the CFRP bars, the samples can be prepared as shown in Figure 3. According to this figure, there are two separate mold systems at both the left- and right-hand sides. The left one is for the preparation of the 18 cm long left part of the samples, whereas the right one is needed for the production of the right 3 cm twin-parts. Furthermore, there is a rigid cap that needs to be connected to the rightmost part of the twin mold system. To mention all the steps to produce the samples, first, the produced CFRP bars are placed as passing through the mold systems. Second, the required amount (see Figure 3) of low-density polyethylene (LDPE) plastic material in granular form is dropped into the molds. With this step, the CFRP bars will be embedded in the plastic materials. Third, the mold systems are given into an industrial oven, which can heat up to the melting point of the plastics, with all materials inside.

### 2.2. Pull-Out Tests

The bonding performance of the CFRP bars in bulk plastics corresponding to the produced samples is performed under pull out tests. The test fixture addressed in this study was developed with the inspiration of the test methodology of the ASTM (D7205/7205M-06) standard due to the absence of a pull-out test methodology belonging to the bonding strength of knotted CFRP bars to bulk plastics in the literature. In other words, the test fixture of this study is developed by utilizing the recommendation of the ASTM standard on supplying an anchor system to be prevented sliding of CFRP bars in anchor filling polymer materials. The detailed demonstration of the developed test fixture can be seen in Figure 4. This test fixture was applied on each sample using the universal test machine of INSTRON-8810 with a load cell capacity of 100 kN. The testing machine supplies the loads corresponding to the pull-out tests, and the displacement data obtained from the testing machine include the combination of sliding and elongations of CFRP bars and the polymer material. Although the loading data of the testing machine can be reliably used for force vs. slip plots, the displacement data of the machine should not be used for the slipping part of the plots. Therefore, as in Figure 4, the usage of LVDT tools were needed to record the slipping data of the CFRP bars in the bulk plastic material. Then, all the data, obtained from the testing machine and the LVDT tools, are synchronously recorded during the pull-out tests in order to reach the force vs. slipping plots. Additionally, the slipping values taken from the LVDT tools are the sliding of the CFRP bars through the bulk plastic material having the critical embedded length indicated in the upper part of the test fixture in Figure 4. This embedded length is 3 cm for each sample as shown in Figure 2, Figure 3 and Figure 4.

To make a detailed description of the other items in Figure 4, the Instron machine has lower and upper jaws in order to apply tensile forces on the test fixture. The lower holders in the test fixture hold the 18 cm long plastic part of the samples, whereas the upper holders are for the 3 cm long plastic part, whose length is demonstrated as “critical embedded length” (CEL.). Additionally, the rigid cap, which is demonstrated in Figure 3 and shown in Figure 4, is connected to the other 3 cm long plastic part for each sample. The main reason for the usage of this rigid cap is to more accurately obtain the slipping data of the CFRP bars integrated with the plastic part and the rigid cap. In other words, while the lower and the upper jaws apply the tensile forces in the test fixture, the integrated ternary materials (rigid cap, bulk plastic and CFRP bar) at the free end of the samples can slide together as a whole. Therefore, the slipping data obtained from the rigid cap is the same with the sliding of the CFRP bars for the samples.

## 3. Results and Discussion

Each sample set mentioned in this study was repeated three times in order to give information about the consistency of the experimental data. Therefore, a total of 18 samples were produced as a result of repeating six different sample sets in three times. The bonding strength of the CFRP bars in the bulk plastics was experimentally revealed by the pull-out tests subjected to the samples. For this reason, in these experiments, the slipping data of CFRP rods inside the bulk plastics and the corresponding pull-out forces were obtained. Figure 4 and Figure 5 show the pull-out forces (y axis) and the slipping data (x axis) obtained from the experimental results of the samples. In addition, Figure 5 deals with the effects of the structural forms (e.g., with or without knots) of the surfaces of CFRP bars on the bonding strength, whereas Figure 6 demonstrates the effects of the size of the cross-sectional area of knotted CFRP bars on the bonding strength.

Figure 5 shows the test results obtained by repeating the samples in three times belonging to the first, second, and sixth sample sets containing different surface forms of the CFRP bars with equivalent diameter thickness. According to these results, it is observed that the bonding strength of the bar type (designated to be LM-FS-8B-CFRP Bars), used in the second sample set, in the bulk plastic material is lower when comparing with the other bar types in Figure 5. Although the bonding strength of the CFRP bars in the bulk plastic material can be slightly increased using the other bar type (designated to be CA-SS-CFRP Bars), the sliding behavior of these bar types in the bulk plastic becomes more fragile (see Figure 5). Therefore, for industry, it does not suggest using this bar type in the bulk plastics. The results at the bottom row in Figure 5 represent the data regarding another bar type (designated to be LM-K-8B-CFRP Bars) used in the sixth sample set. According to these results, the usage of this CFRP bar type in the samples causes that the interfacial bonding strength between the bars and the plastic material increases dramatically and the structural system of the samples is also enough ductile for the industry. In other words, the usage of the knotted CFRP bars increases the friction and mechanical interlocking on adhesion strength of CFRP–plastics interfaces.

Figure 6 addresses the size effect of the cross-sectional area of the knotted CFRP bars on the bonding performance in the low-density polyethylene plastic material. Therefore, Figure 6 shows the test results obtained by repeating the samples in three times belonging to the third, fourth, fifth, and sixth sample sets consisting of knotted CFRP bars with different sizes. According to these results, the bonding strength of the CFRP bars in the plastic material increases with increasing size of the cross-sectional areas. In addition, it is observed that the structural system behavior (pull-out force vs. slipping) of all the samples addressed in Figure 6 is ductile due to the usage of the knotted CFRP bars. Thus, for industry, it may be suggested using knotted CFRP bars in bulk plastics to have more ductile and strengthened structural composite systems.

Table 2 presents a summary for the test results of the sample sets by containing the ultimate pull-out force values of the plots in Figure 4 and Figure 5. When these values are examined, it shows that the bonding strength of the CFRP bars in the plastic material can be increased by 233% forming knots on the bars. Table 2 also shows the effect of increasing the cross-sectional size of the knotted bars on the bonding strength with numerical values. In conclusion, all the data in Table 2 indicate that the knotted and thicker reinforcements will adhere better to plastics. In this context, it is revealed with this study that engineers should take account of these details when reinforcing plastics.

It is thought that reinforced plastics can be used more commonly by industries with the improvement of the interfacial bonding strength between the plastic and reinforcing materials. Therefore, the fact that the innovations discussed in this study can positively affect the composite industry shows that this study has a great value. In other words, it is thought that the plastic structural systems, which are reinforced by the knotted bars, will be more preferred by the industry in near future due to their increasing strength.

Figure 7 is a representation to examine the interaction mechanism of the bonding strength of different fiber reinforcement types, which can be used in bulk plastics. In this figure, the surface deformation depth created on these fiber reinforcement types is illustrated. According to Figure 7, the knotted carbon-based reinforcements discussed in this study have the highest surface deformation depth among the other reinforcement types, which are commonly used in concrete material. For example, one of the samples (LM-K-8B-CFRP) in this study has the depth ratio to the diameter of the fiber reinforcement to be 0.41, whereas, in a previous study cited in the literature [39], this ratio of a ribbed reinforcement sample is 0.06. When these different fiber reinforcement types are used to strengthen bulk plastics, it is thought that the knotted reinforcement with higher surface deformation depth will have higher bonding strength against slipping compared to the others because of the fact that plastic material can be easily deformed under loading. In other words, the plastic material can easily allow the movement of the fiber reinforcements, which are subjected to tensile or compression loads, by creating an expanded space path around the reinforcements due to its low stiffness property. Considering that the passage of the reinforcements in this expanded path is determined by the size of the surface deformation depth illustrated in Figure 7, it is expected that the knotted reinforcement addressed in this study will have more resistance to slip compared to the others. Additionally, the bonding strength of the reinforcement types also depends on the size of this expanded space path around the reinforcements. Therefore, when any type of fiber reinforcement is used to strengthen a different material such as plastic or concrete, the size of the expanded path will affect the bonding strength of the fiber reinforcement. In other words, when a fiber reinforcement is used in concrete material, the size of the expanded path will be smaller than to be used in plastic, because the concrete has both a lower Poisson ratio and a higher stiffness under compression force compared to the plastic material. In this respect, it is predicted that the importance of surface deformation depth will decrease partially when fiber reinforcements are used in concrete material. In this case, perhaps the bonding strength of the fiber reinforcements will be determined by the shear capacity of the surface deformation zones rather than the size of the surface deformation depth.

According to Table 2, it can be stated that the knotted surface structure was able to increase the bonding capacity of the CFRP bars in plastic material by 233% (by the comparison of the sample sets of CA-SS-CFRP and LM-K-8B-CFRP bars with the similar diameter size). Considering some studies in the literature, this increment is not a rate to be taken lightly for the bonding strength of FRP bars. However, it should be also declared that these studies in the literature had been focused on the bonding strength of FRP bars in concrete, not the plastic material. For example, according to the results of the experiments in a recent study [29], CFRP bars with ribbed surface deformation exhibited approximately 22% higher bonding strength in concrete than smooth-surfaced ones.

It is thought that the preparation process is likely to damage some of the fibers at the knotted zone during the preparation of the knotted CFRP bars in this study. Therefore, it is highly probable that the total tensile capacity of the fiber reinforcements has been reduced due to surface stripping while trying to increase the bonding strength of the fiber reinforcements in the bulk plastic material. In this context, although knotted reinforcements are used to strengthen bulk plastic materials, it is important to design these composite structures based on an optimization between the reduced tensile capacity and increased bonding strength of the fiber reinforcements. On the other hand, as carbon fiber reinforcements have already very high tensile capacities, the importance of any contribution to increase the bonding strength of the fiber reinforcements is undeniable for the design of the reinforced composite plastic structures.

It is thought that the practical application of the knotted CFRP bars mentioned in this current study will take place in the aviation, aerospace, energy, and automotive industries. For example, the author has recently worked on new generation wind turbine blades fabricated using knotted CFRP bars in bulk plastic structures, which have the blade aerodynamic geometry in 3D [40]. In addition, the new generation turbine blades produced in this study were compared with commercial blades under mechanical tests. The fact that the new generation blades exhibited a better stiffness performance than the commercial ones will have an effect that will increase the use of the knotted reinforcements mentioned in this current study. Finally, the interval distance between each knot and the improvements on the mass production process of the knotted CRFP bars are recommended to be investigated for future studies.

## 4. Conclusions

As the novelty of this current work, formation of knots was offered for carbon-based reinforcements in order to enhance the interfacial bonding strength of the reinforcements. In order to measure this bonding strength, several sample sets of bulk plastic composites reinforced with the knotted CFRP bars were subjected to pull-out tests. The main observations and conclusions of these test results are summarized as follows:According to the results of the pull-out tests, the interfacial bonding strength between the knotted carbon-based reinforcements and the bulk plastic materials could be increased up to 233%;This study also discussed the effect of the size of the cross-sectional area of the knotted reinforcements on its bonding strength in plastic materials. According to the experimental results, it has been observed that the bonding strength of the reinforcements is proportionally increased with the cross-sectional area. In other words, it was measured that the increment of the bonding strength of the knotted reinforcements, which were produced using 2, 4, 6, and 8 carbon fiber bundles is 27%, 87%, 177%, and 233%, respectively;By comparing the first and second sample sets, although the laboratory-made flat surfaced CFRP bars have slightly less bonding strength than the commercially available smooth-surfaced reinforcements, it was observed that the sliding failure of these lab made CFRP bars was much more ductile;Experimental test results also showed that the sliding of the knotted CFRP bars in the bulk plastic material exhibits more ductile behaviors than the commercially available smooth-surfaced reinforcements;By considering that the ductility of the sliding failures has also importance for the prevention of sudden collapses for composite structures, it can be stated that the knotted CFRP bars have more benefits than the smooth-surfaced reinforcements for composite community.

To conclude, it is thought that the data obtained from the experimental results in this study will increase the interest of various industries to use the knotted reinforcements in composite structures.

## Figures and Tables

**Figure 1 polymers-15-02036-f001:**
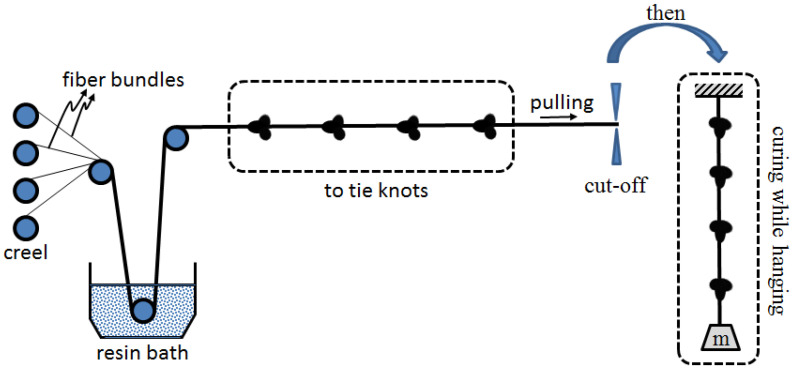
Representative preparation process of a knotted FCRP bar using an improved impregnation system.

**Figure 2 polymers-15-02036-f002:**
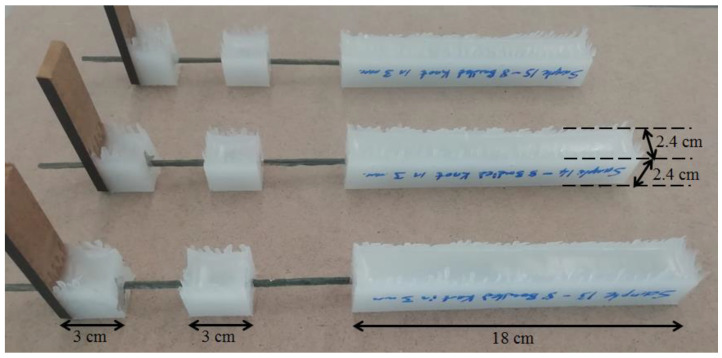
Selected samples after production.

**Figure 3 polymers-15-02036-f003:**
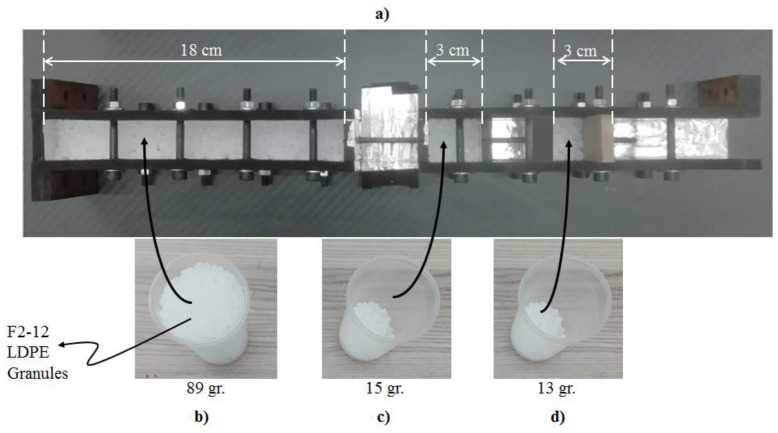
Sample preparation process in a mold system. (**a**) The mold systems; (**b**) 89 gr. F2-12 LDPE granule materials in a pet-cup; (**c**) 15 gr. F2-12 LDPE granule materials in a pet-cup; (**d**) 13 gr. F2-12 LDPE granule materials in a pet-cup.

**Figure 4 polymers-15-02036-f004:**
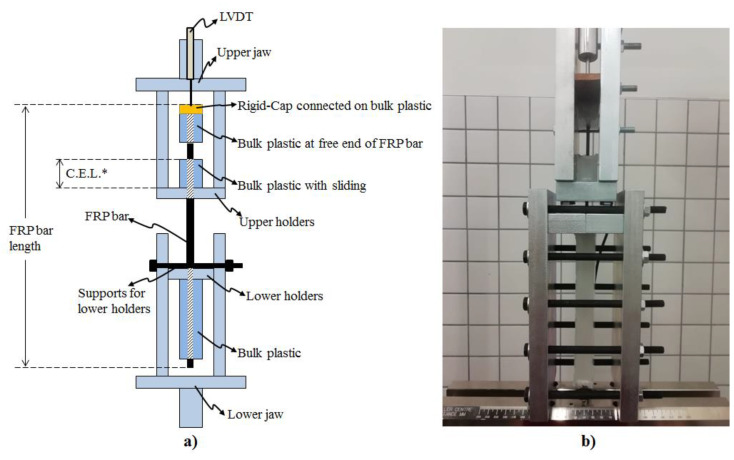
Sample pull-out test fixtures. (**a**) Representative front view for the test fixture; (**b**) real case view for the test fixture in INSTRON mechanical test machine. * C.E.L.: Critical Embedded Length.

**Figure 5 polymers-15-02036-f005:**
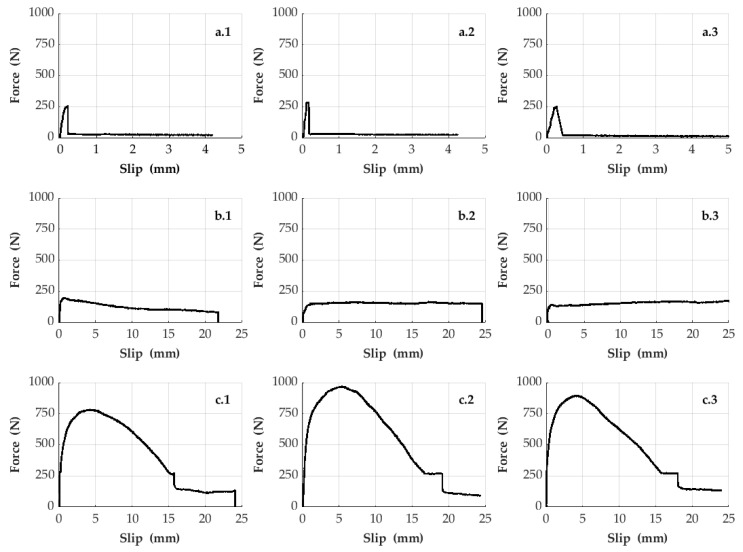
Pull-out test results belonging to the first, second and sixth sample sets. (**a.1**,**a.2**,**a.3**) refer to the three times repeated samples of the first sample set, respectively. (**b.1**,**b.2**,**b.3**) refer to the three times repeated samples of the second sample set, respectively. Finally, (**c.1**,**c.2**,**c.3**) refer to the three times repeated samples of the sixth sample set, respectively.

**Figure 6 polymers-15-02036-f006:**
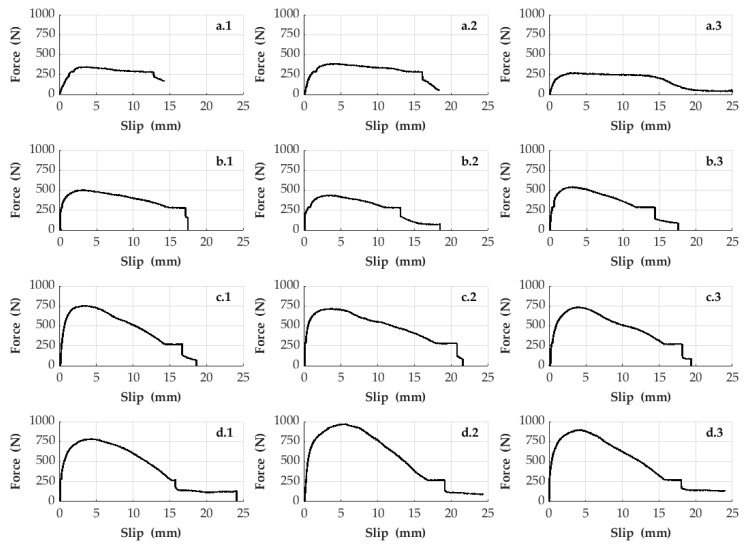
Pull-out test results belonging to the third, fourth, fifth and sixth sample sets. (**a.1**,**a.2**,**a.3**) refer to the three times repeated samples of the third sample set, respectively. (**b.1**,**b.2**,**b.3**) refer to the three times repeated samples of the fourth sample set, respectively. (**c.1**,**c.2**,**c.3**) refer to the three times repeated samples of the fifth sample set, respectively. Finally, (**d.1**,**d.2**,**d.3**) refer to the three times repeated samples of the sixth sample set, respectively.

**Figure 7 polymers-15-02036-f007:**
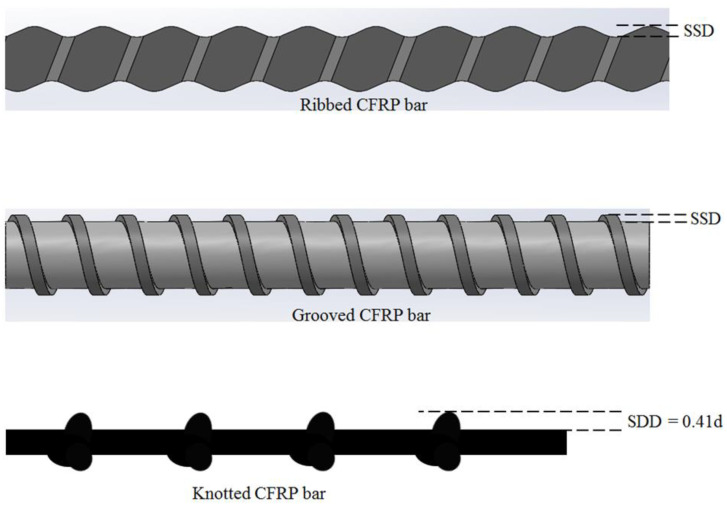
Surface deformation depth (SSD) representations for selected fiber reinforcement types used in the literature and this study. The ratio (0.41) of SSD to d (diameter of the reinforcement) were calculated for the knotted sample (LM-K-8B-CFRP) by measuring. Additionally, it should be noted that this calculated ratio varies depending on the number of fiber bundles used to create the fiber reinforcement.

**Table 1 polymers-15-02036-t001:** Designations and views for the sample sets.

Designations for the Sample Sets	Reinforcing Bars Embedded in the Sample Sets	Representative Figure for the Sample Sets
CA-SS-CFRP	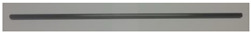	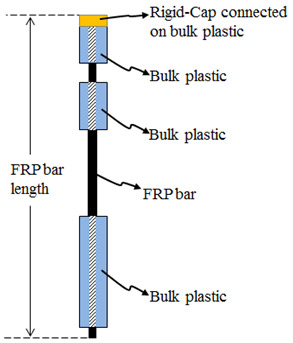
LM-FS-8B-CFRP	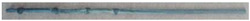
LM-K-2B-CFRP	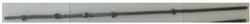
LM-K-4B-CFRP	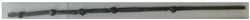
LM-K-6B-CFRP	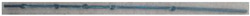
LM-K-8B-CFRP	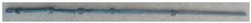

**Table 2 polymers-15-02036-t002:** Ultimate sliding forces regarding the pull-out test results.

Bar Types Used in the Sample Sets	Sample No.	Ultimate Slip Force (N)	Average Slip Force (N)	Normalized Average Slip Force (%)
CA-SS-CFRP Bars	1	257.91	266.97	
	2	288.72		100
	3	254.27		
LM-FS-8B-CFRP Bars	1	203.08	184.45	
	2	169.44		69
	3	180.83		
LM-K-2B-CFRP Bars	1	349.66	338.43	
	2	389.50		127
	3	276.14		
LM-K-4B-CFRP Bars	1	508.54	499.38	
	2	442.77		187
	3	546.84		
LM-K-6B-CFRP Bars	1	761.32	739.97	
	2	718.86		277
	3	739.72		
LM-K-8B-CFRP Bars	1	787.96	888.29	
	2	976.07		333
	3	900.83		

## Data Availability

The data presented in this study are available on request from the corresponding author.

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
