# Peer review of "Experimental Investigation on the Bonding Strength of Knotted CFRP Bars in Bulk Plastics"

_polymers, 2023, doi:10.3390/polym15092036_

Round 1

Reviewer 1 Report

The knotted FRP bars were designed to improve the interface bonding strength in bulk plastics. The ideas of this paper are innovative, which also include some interesting results. However, the results and analysis should be further enriched through considering the following comments. 

1. Please provide some quantitative analysis results in the abstract. In addition, the interaction mechanism of bonding strength for different CFRPs with bulk plastic should be further exposed and revealed.

2. In addition to robust specific strength and stiffness, the main factors to be considered in the engineering application of carbon fiber reinforced polymer composites also include excellent corrosion resistance, fatigue resistance and creep resistances compare to the other synthetic fiber reinforced polymer composites (such as GFRP and BFRP). The following latest relevant research can be further referenced to fill this gap. Composite Structures, 2022, 293, 115719. Construction and Building Materials, 2019, 206: 279-286. Engineering Structures, 2023, 274: 115176.

3. What is the difference of interface bonding between the carbon fiber or glass fiber composite in concrete and in bulk plastics? Please provide relevant explanations.

4. In this study, it is recommended to create knots on the reinforcements to make them difficult to slip in the bulk plastics. What is the effect of knots on the reinforcements on the mechanical properties? In addition, how is the complexity of the preparation process?

5. Please provide some basic information about raw materials, such as performance, size, manufacturer source, etc.

6. During the preparation of knotted FRP bars, whether the preparation process damage the surface of FRP, such as the surface stripping?

7. In the results and discussion section, please provide the relevant data of interface bonding strength for the different FRPs. In addition, the interface damage mode and interface interaction mechanism should be further provided and analyzed.

8. The clarity of Figure 4 and Figure 5 is not enough, and each picture is too small. It is recommended to make corresponding adjustments.

9. Conclusions should be considered to include the important information and findings related to this paper.

Reviewer 2 Report

This paper aims to explore the bonding strength of the reinforcements in bulk plastics using knotted CFRP bars. Following are my comments that must be addressed in the revised manuscript:

1.      Modify the title.

2.      Abstract need to be rewritten. Start the Abstract with the significance of the research, followed by methods, conclusions.

3.      There is no logic in the introduction section; it has many paragraphs. Please improve and add more literature. Please consider the other studies as well, e.g. https://doi.org/10.1016/j.conbuildmat.2020.121766 ; https://doi.org/10.1016/j.compositesb.2021.109219 ;

4.      The significance and novelty of the research need to be added in the last paragraph of the introduction section. What is the scientific contribution of this research?

5.      A new section before the conclusion should be added to describe the main findings of the research and compare them with the previous studies. Also, discuss the practical application of current work.

6.      Report advantages and disadvantages of bond strength.

7.      Please add quantitative results to the conclusions.

8.      What are your future recommendations.

9.      The Figure's quality is low, i.e. Figure 4 and 5.

10.  The conclusions are not technically written. It is suggested to write the conclusions points in a scientific way.

Round 2

Reviewer 1 Report

It is suggested to accept the paper.

Reviewer 2 Report

Most of my comments have been addressed. However, conclusions need to be improved and written technically. Start the conclusion section with an introductory paragraph describing the methods employed, and write the main conclusions as bullet points. Minimum 5 points should be written in the conclusion section
